# Inositol trisphosphate and ryanodine receptor signaling distinctly regulate neurite pathfinding in response to engineered micropatterned surfaces

Joseph T. Vecchi [1,2], Madeline Rhomberg[2], C. Allan Guymon[3], Marlan R. Hansen [1,2]*

1 Department of Molecular Physiology and Biophysics, Carver College of Medicine, Iowa City, IA, United States of America, 2 Department of Otolaryngology Head-Neck Surgery, Carver College of Medicine, Iowa City, IA, United States of America, 3 Department of Chemical Engineering, University of Iowa, Iowa City, IA, United States of America

* marlan-hansen@uiowa.edu

**Data Availability Statement:** All relevant data are within the manuscript and its Supporting Information files.

## Abstract

Micro and nanoscale patterning of surface features and biochemical cues have emerged as tools to precisely direct neurite growth into close proximity with next generation neural prosthesis electrodes. Biophysical cues can exert greater influence on neurite pathfinding compared to the more well studied biochemical cues; yet the signaling events underlying the ability of growth cones to respond to these microfeatures remain obscure. Intracellular $Ca^{2+}$ signaling plays a critical role in how a growth cone senses and grows in response to various cues (biophysical features, repulsive peptides, chemo-attractive gradients). Here, we investigate the role of inositol triphosphate (IP3) and ryanodine-sensitive receptor (RyR) signaling as sensory neurons (spiral ganglion neurons, SGNs, and dorsal root ganglion neurons, DRGNs) pathfind in response to micropatterned substrates of varied geometries. We find that IP3 and RyR signaling act in the growth cone as they navigate biophysical cues and enable proper guidance to biophysical, chemo-permissive, and chemo-repulsive micropatterns. In response to complex micropatterned geometries, RyR signaling appears to halt growth in response to both topographical features and chemo-repulsive cues. IP3 signaling appears to play a more complex role, as growth cones appear to sense the microfeatures in the presence of xestospongin C but are unable to coordinate turning in response to them. Overall, key $Ca^{2+}$ signaling elements, IP3 and RyR, are found to be essential for SGNs to pathfind in response to engineered biophysical and biochemical cues. These findings inform efforts to precisely guide neurite regeneration for improved neural prosthesis function, including cochlear implants.

## Introduction

The intricate process of neurite growth and guidance is fundamental for the establishment of functional neuronal circuits, both during neurodevelopment and in regenerative processes.

**Funding:** MH and AG: NIH-NIDCD R01-DC012578 JV: NIH-NIDCD F31-DC020371, and University of Iowa: NIGMS T32-GM007337.

**Competing interests:** The authors have declared that no competing interests exist.

**Abbreviations:** BDNF, Brain-derived neurotrophic factor; CI, Cochlear implant; DRGN, Dorsal root ganglion neuron; FBS, Fetal bovine serum; HBSS, Hanks' balanced salt solution; HDDMA, 1,6-hexanediol dimethacrylate; HMA, hexyl methacrylate; IP3, inositol tripohosphate; NF200, High molecular weight neurofilament protein; NT-3, Neurotrophin-3; PBS, Phosphate buffered saline; RyR, Ryanodine receptor; SGN, spiral ganglion neuron..

The precise targeting of axons and dendrites is determined by the growth cone integrating and responding to diverse biochemical and biophysical gradients and borders as it is pathfinding [1, 2]. Studying this process is essential to understand neural development as well as to engineer de novo neurite growth towards desired targets such as neural electrodes. Precisely directing neurite growth towards stimulating or recording electrodes has the potential to significantly enhance the performance of neural prosthetics [3, 4]. In response to these goals, crafting specific patterns of guidance cues to precisely direct neurite growth has been an active area of research [5, 6].

Recent advancements have introduced micro and nanoscale biochemical and topographical features as tools to influence neurite growth and neural circuitry [7, 8]. In this, the growth cone plays a pivotal role in pathfinding by responding to diverse environmental signals, including biochemical gradients, matrix proteins, and biophysical cues. Understanding how the growth cone integrates these cues is crucial for enhancing neural-biomaterial interactions and designing effective neural prosthetics. The role of $Ca^{2+}$ signaling in growth cone guidance has been well-established, with localized concentration gradients generated within the growth cone to orchestrate turning responses [9, 10]. Interestingly, recent evidence suggests that the intracellular signaling mechanisms which govern neurite guidance in response to biochemical cues also play a role in topographical neurite guidance [11, 12]. However, the mechanisms through which growth cones transduce biophysical cues into directed neurite growth, and if these mechanisms are generally recruited in response to other cues, remain largely unexplored.

Spiral ganglion neurons (SGNs), the target neurons of cochlear implants, offer a valuable model for studying neurite alignment and turning in response to biophysical and biochemical cues. Studying these neurons is important since stimulation and precise guidance of neurite outgrowth into close proximity stands to enhance the functionality of next generation cochlear implants (CIs). In SGNs and other sensory neurons, the importance of $Ca^{2+}$ signaling, and inositol triphosphate (IP3) and ryanodine receptors (RyR) in particular, for pathfinding has been established [13, 14]. Multiple studies demonstrate contribution of IP3 signaling in growth cone steering, especially in turning in response to chemo-attractive cues [15–17] while the role of RyR signaling appears more complex [18]. RyR signaling does not appear to contribute to turning behavior in response to chemo-attractive cues [15], but is necessary for neurite alignment to biophysical cues [17] and may be involved in regenerative outgrowth or degeneration in certain contexts [19–21].

Therefore, while intracellular $Ca^{2+}$ signaling events have been implicated in neurite guidance, integrating these findings into a comprehensive model of neurite pathfinding in response to various cues is needed. Neurite pathfinding represents a multi-step process, wherein a growth cone must effectively perceive cues of different types (soluble biochemical, substrate proteins, biophysical), synthesize and integrate the signals, and subsequently execute an appropriate behavioral response (repulsion, turning, growth). It is clear that both IP3 and RyR signaling are involved in neurite pathfinding, however, further investigation is warranted to elucidate the general role intracellular $Ca^{2+}$ signals play in governing how a growth cone responds to diverse cue types and geometries. In particular, focus should be given to the specific aspects (repulsion, turning, growth) of neurite pathfinding that these signaling pathways regulate as well as the specific growth cues which activate IP3R and RyR. Consequently, this study seeks to establish parallels between the intracellular signaling events that are triggered by biochemical and topographical cues, therefore aiming to identify master regulators that underlie the complex processes of neurite sensing and pathfinding in response to growth cues.

To investigate the intricate processes of neurite growth, guidance, and signal transduction across various guidance cues, we engineered various, novel micropatterned surfaces and

analyzed the role of intracellular $Ca^{2+}$ signals on neurite pathfinding in response to the diverse cues using targeted pharmacologic inhibitors. These engineered substrates include three different topographically micropatterned systems and two biochemically patterned surfaces. The substrates all consist of stable growth cues and represent a wide array of cues presented to the growing neurite. In particular, we engineer the substrates to consist of cues ranging from, chemo-permissive, chemo-repulsive, repeating linear topographies, repeating multidirectional topographies, and topographical turn challenges. In studying neurite growth in response to these substrates, we find that IP3 and RyR signaling each contribute to growth cone responses to these varied biophysical and biochemical cues, however in distinct ways, suggesting generalized mechanisms regulating neurite pathfinding. Our findings inform fundamental processes governing neural regeneration as well as the design of advanced neural prosthetics that can integrate with target neural tissues to more accurately recapitulate the intimate circuitry of the native nervous system.

## Methods

### Animals

All procedures involving animals were conducted in accordance with the NIH Guide for the Care and Use of Laboratory Animals and were approved by the University of Iowa Institutional Animal Care and Use Committee via written consent of the protocol, #1101569. All mice were maintained on a C57BL/6 (Envigo) background and housed in groups on a standard 12:12 h light: dark cycle with food and water provided ad libitum. Mice were sacrificed for experimental use at p3-5 of age without anesthesia via rapid decapitation with sharp scissors. This method was chosen to rapidly sacrifice the animals and minimize neural damage. Importantly, SGN peripheral processes already contact hair cells in the organ of Corti at age p3-5. For the live cell imaging, neurons were harvested from Pirt-GCaMP3 transgenic mice gifted from Dr. Xinzhong Dong of John Hopkins University [22].

### Micropatterned topographical substrates

Topographically micropatterned substrates were generated as previously described using photopolymerization [23]. First, a monomer solution was made consisting of 40 wt% hexyl methacrylate (HMA, Aldrich), 59 wt% 1,6-hexanediol dimethacrylate (HDDMA, Aldrich), and 1 wt% of 2,2-dimethoxy-2-phenylacetophenone (DMPA, BASF). Then, to create the patterned substrates, this solution was evenly dispersed on a silane-coupled piece of cover glass by placing the glass-chrome custom photomask (Applied Image Inc.) on top. These samples were then exposed to 365 nm light at an intensity of $16mW/cm^2$ using a high-pressure mercury vapor arc lamp (Omnicure S1500, Lumen Dynamics, Ontario, Canada) to polymerize the monomer solution. The opaque portions mask areas of monomer from UV radiation thereby modulating the rate of the polymerization locally to generate features on the surface. The mask induced light modulation creates differential polymerization regions with raised features or ridges underneath transparent bands where UV light intensity and the polymerization rate are highest. These substrates direct the growth of various cells and neurons, including SGNs and DRGNs [24]. Additionally, various geometries of topographical substrates can be generated by varying the photomask [25] or UV light exposure (to change feature amplitude) [23]. In such, here we utilize 3 different geometries for the micropatterned features. First a pattern of narrow repeating rows of ridges and grooves with a periodicity of 10 μm and amplitude of 3 μm [26]. Second is a pattern of repeating 90° zigzags with a periodicity of 50 μm and amplitude of 4 μm [25]. Lastly, we created a novel system with 6 different patterned turn challenges, ranging from

30 to 180 (straight) degrees, where the width of the microfeature groove is 20 µm and amplitude is 4 µm.

The dimensions and characteristics of every substrate used in the following experiments was measured using white light interferometry (Dektak Wyko 1100 Optical Profiling System, Veeco) [23]. Feature amplitude was characterized in 5 regions of the glass coverslip and neurons were only cultured on areas within +/- 10% of the target microfeature amplitude.

## Creation of chemo-permissive laminin stripe pattern

Unpatterned HDDMA/HMA substrates were biochemically patterned with laminin stripes as previously described [27]. Briefly, a solution of laminin (40µg/ml, Sigma-Aldrich) with 0.05 wt % 2-hydroxy-1-[4-(2-hydroxyethoxy)phenyl]-2-methyl-1-propanone (HEPK, Ciba), a photo-radical generator, was pipetted onto these acrylate substrates. A photomask with repeating 25µm stripes and clear bands was placed on the substrate to evenly distribute the liquid. The sample was illuminated for 50 s with UV-light using a mercury vapor arc lamp (Omnicure S1500) at intensity of 16 mW/cm$^2$ as measured at 365 nm to denature and deactivate the protein exposed to light. The photomask was then removed and the surface was washed once with PBS(-/-) and stored in PBS(-/-) until use.

## Creation of chemo-repulsive EphA4-Fc stripe pattern

Unpatterned HDDMA/HMA substrates were biochemically patterned with EphA4-Fc stripes on a uniform laminin coating as previously described [28]. First, substrates were uniformly coated by evenly distributing a laminin solution (20µg/ml, Sigma-Aldrich) overnight at 4˚C. Next, a topographical micropattern with alternating ridges and troughs defined by a periodicity of 50 µm and amplitude of 2 µm was uniformly coated with EphA4-Fc (10 µg/ml, R&D Systems) for 1 h to create a peptide "stamp". After removing the excess solution from both the stamp and the laminin substrate, the stamp was firmly pressed onto the laminin substrate to create the EphA4-Fc stripe pattern.

Immunofluorescent labeling was incorporated in each experiment to confirm the presence of the peptide pattern. For this, primary and secondary antibodies were added to the labeling procedure as described in more detail in the section "Comparing rDRGN growth cone morphology on micropatterned substrate" below. Specifically, anti-laminin rabbit polyclonal antibody (1:500, Abcam) was added followed by Alexa 633 conjugated secondary antibody (1:800, ThermoFisher). Since EphA4-Fc is a chimeric peptide, only labeling with Alexa 488 conjugated anti-human IgG antibody (1:800, ThermoFisher) is required to detect and label the Fc portion of the peptide. EphA4-Fc functions as a repulsive growth cue to SGN neurites [28–32].

## Replated dorsal root ganglion neuron cultures

DRGNs were isolated as previously described, [33] however, using neonatal (p3-p5) mice. First, a 24-well polystyrene plate was coated with poly-L-ornithine solution (Sigma-Aldrich) for 1 h at RT. The surface was washed with sterile Milli-Q® three times prior to a laminin solution (20 µg/ml, Sigma-Aldrich) being applied and incubated overnight at 4˚C. After warming the coated well plate for 1 h at 37˚C, the freshly dissected DRGNs were cultured on this plate for 72 h before being replated. The replating procedure has been described previously, [34] but after a 1 min incubation with TrypLE Express (Thermo Fisher), warm media was used to gently triturate the culture surface and lift the adhered neurons. The resulting replated DRGNs (rDRGNs) were then cultured on the micropatterned or unpatterned HDDMA/HMA substrates in a humidified incubator with 6% CO$_2$. Each experiment using rDRGNs was

repeated three times, where each experimental replicate was conducted utilizing a pool of DRGs from the lumbar region of one mouse.

## Spiral ganglion neuron cultures

The day before the biochemical patterning and culture, substrates were sterilized by soaking in 70% ethanol for 2 min and then exposure to UV light in a cell culture hood for 15 min. After creating the biochemical patterns of interest cloning cylinders were placed on the patterned coverslips. SGN cultures were prepared from mice between post-natal day 3 and 5 (p3-p5). Mice were decapitated, and their cochleae were isolated from their temporal bones in ice-cold PBS. Then the spiral ganglia were isolated and placed into ice-cold HBSS(-/-) as previously described. [35] Enzymatic dissociation was then performed in calcium- and magnesium-free HBSS with 0.1% collagenase and 0.125% trypsin at 37°C for 25 min. 100 μL of FBS was added to stop dissociation. Ganglia were washed with Neurobasal media before placing in supplemented Neurobasal Medium (Thermo Fisher) containing: 5% FBS, 2% N2 Supplement (Thermo Fisher), 10 μg/mL Insulin, 50 ng/mL BDNF (R&D Systems), and 50 ng/mL NT-3 (Sigma-Aldrich). Once in supplemented media, ganglia were triturated first with a 1000 μL pipette tip followed by a 200 μL tip. Cultures were plated onto micropatterned substrates and maintained in a humidified incubator with 6% $CO_2$ for 48 h. Every experiment utilized this method of deriving a neuronal culture from a pool of spiral ganglia harvested from multiple litter mates. Importantly, while the culture for each experiment was repeated three independent times, the sample size used for statistical analysis for all experiments were the neurons measured in each assessment.

## Targeted inhibition of $Ca^{2+}$ signaling

To selectively modulate $Ca^{2+}$ signaling, 100 μM ryanodine (Abcam) was used to inhibit RyR signaling and 2 μM xestospongin C (Cayman Chemical) to inhibit IP3 signaling. The drugs and concentrations were chosen based on their prior in studying neurite guidance. 100 μM ryanodine has been shown to trap RyRs in a closed state while not effecting SGN viability [15, 17]. 2 μM xestospongin C has been shown to nearly completely block IP3 transport, while also not effecting SGN viability [17, 36].

## Calcium imaging

To study the morphology and behavior of neuron growth cones in response to topographical growth cues, rDRGNs from mice expressing Pirt-GCaMP3, a genetically encoded calcium indicator [22, 37], were cultured on a topographically patterned substrate consisting of repeated rows of ridges and grooves (10 μm periodicity and 3 μm amplitude) and compared against unpatterned substrates. The substrates were made in the same manner as described previously [23].

After overnight incubation, half the media was removed and fresh media containing drug was added. After 1 h incubation with the inhibitor, the sample was placed in an Okolab incubator and imaged with 40x objective using a Lecia DMIRE2 equipped with a FITC/EGFP filter (EX 480nm +/- 20nm, EM 535nm +/- 25nm, Chroma Technology) and a custom filter (EX 405nm +/- 10nm, EM 520nm +/- 20nm, Chroma Technology). Importantly, the Pirt-GCaMP3 enables fluorescent visualization of the entire rDRGN, including the growth cone. In each sample, five growth cones were randomly selected to be imaged. Each growth cone was imaged by three images taken 10 s apart by the two wavelengths.

After images were gathered, relative $Ca^{2+}$ in the growth cone was determined as follows. First, two signals were averaged from the three images; both from a round region of interest

(ROI) which filled the growth cone of interest as much as possible and an ROI from the background adjacent to it. After correcting each wavelength to the background signal, the 535nm (FITC/EGFP) signal was divided by the 520nm signal to find a measure of the relative $Ca^{2+}$ level. For each experimental replicate, the data was normalized to the level of the untreated neurons cultured on an unpatterned substrate. Since $Ca^{2+}$ level was compared in response to two potentially interacting variables, substrate and drug treatment, two-way ANOVA was used to assess significance.

## Comparing rDRGN growth cone morphology on micropatterned substrate

For the rDRGNs, chicken polyclonal antibody to NF200 (RRID: AB_2313552, Aves) was added (1:800 in blocking buffer) for 1 h at RT. Following this incubation, the culture surface was washed three times with PBS (-/-) and then Goat anti-Chicken Alexa Fluor®488 (RRID: AB_2534096, ThermoFisher) and Alexa Fluor™ 546 Phalloidin (ThermoFisher) (1:200) in blocking buffer (1:1000) was added for 1 h at RT in the dark. Following the secondary antibody incubation, cultures were again washed with PBS (-/-) three times. Lastly, the polymer-coated cover glass was mounted onto a glass slide using Fluoromount-G® (SouthernBiotech) and kept in the dark for 24 h before imaging.

Fixed growth cones were imaged via confocal microscopy (STELLARIS 8, Leica). Images of the growth cones were analyzed by two approaches using Imaris software. First, the shape of the growth cone was computed, in particular, the prolate ellipticity. For this, the software approximated the shape of the growth cone as an ellipsoid and prolate ellipticity was derived as indicated by the equation. Second, the 2D shape of the growth cone was approximated as an ellipse and its major axis was derived. The angle made between this major axis and the neurite shaft was measured to the nearest 5˚ (Fig 2).

## Immunofluorescent labeling of SGNs

Media was removed and the culture surface was washed three times with phosphate-buffered saline without calcium and magnesium (PBS(-/-)) and 4% paraformaldehyde in PBS(-/-) (Fisher Scientific) was added to fix the cultures. After 20 min at RT, cells were washed with PBS(-/-) three times, and then blocking buffer was added (5% normal goat serum [Thermo-Fisher], 0.2% Triton™ X-100 [Fisher Scientific], and 1% BSA [Research Products International] in PBS[-/-]). After incubating for 30 min at RT mouse monoclonal anti-NF200 antibody (RRID: AB_260781, Millipore Sigma) was added (1:400 in blocking buffer) for 2 h at 37˚C. Following this incubation, the culture surface was washed 3 times with PBS (-/-) and then Goat anti-mouse Alexa Fluor®546 (RRID: AB_2534089, ThermoFisher) in blocking buffer (1:800) was added for 1 h at RT in the dark.

SGNs were grown on these substrates, exposed to the same inhibitors of $Ca^{2+}$ mentioned above, and their alignment to the stripes was quantified. It is important to note that prior work has shown no effect on viability of neurons grown on these polymer substrates [26]. Neurite alignment to the peptide stripes was quantified by taking the total length of neurite growth and dividing it by the proportion of that length which is in the direction of the pattern [26, 38]. Thus, 1 would indicate perfect alignment while 1.5 would represent a random orientation of growth. This alignment index was also measured for SGNs growing on the EphA4-Fc stripes as well. Kruskal-Wallis was used to compare alignment in both experimental settings.

## Quantifying SGN neurite growth in zigzag pattern

Multidirectional topographical substrates were first coated with poly-L-ornithine solution (Sigma-Aldrich) for 1 h at RT. After washing the substrate with $H_2O$, a laminin solution was

added (20µg/ml, Sigma-Aldrich) and incubated overnight at 4˚C. SGNs grown on these substrates were traced and a variety of measurements were derived. Full neurite tracing was conducted using NeuronJ [39] and data were analyzed via a previously described MATLAB program [25, 28]. First, the neurite tracing is divided into 10 µm length segments and the angle relative to the horizontal direction was calculated, as previously described [27]. Of note, overall, the zigzags run in the horizontal direction and each feature segment is aligned at 45˚. Using these angle segments, the number of turns for each neurite is determined. A turn was designated by the MATLAB program if the neurite demonstrated a trajectory difference of at least 10˚ for three consecutive segments from the previous three consecutive segments [25]. Additionally, the percent of neurite length in the trough of the topographically patterned substrate was determined [25, 28].

## Semi-automated measurement of SGN neurite length within microfeature

Another method for assessing neurite pathfinding behavior was studying neurite pathfinding in response to the micropatterned substrate which consisted of six different turn challenges. A first measurement assessed every SGN neurite that encountered a micropatterned, topographical microfeature by measuring the neurite length in the microfeature using NeuronJ [39, 40]. The neurite length in microfeatures was measured from where the neurite entered the microfeature to where it either exited the microfeature or the endpoint of the neurite. These length data were then grouped both by turn angle and drug treatment. For this approach, the unpatterned control condition was generated by overlaying the shape of the 60˚ microfeature over a flat portion of the substrate, and length data was measured in the same manner as neurites encountered this pseudo-pattern. Two-way ANOVA was used to statistically assess these data since SGN neurite behavior was compared in response to two potentially interacting variables, turn angle and drug treatment.

## Manual assessment of SGN neurite pathfinding

Another method for assessing neurite pathfinding behavior was qualitatively scoring every neurite encounter with one of the five angled microfeature turn challenges on the micropatterned substrate [40]. Neurites were scored as either turning or not turning. In particular, two conditions in which neurites were scored as not turning included: 1. not following the microfeature; neurite growth not orienting in the direction of the pattern and the neurite crossing the microfeature in the direction approximately perpendicular to the microfeature direction, and 2. neurite following the straight portion of the microfeature but failing to complete the turn challenge and crossing the microfeature. Additionally, two categories of neurites were deemed to successfully turn in response to the microfeatures, as previously described [40], including: 1. neurite following microfeature and navigating one turn; the neurite entering the microfeature more than 30 µm before the turning challenge and remaining positioned in the microfeature 30 µm past the vertex of the turning challenge, as well as 2. neurite navigating multiple turns and/or the shaft reorienting where it is aligned across multiple of the zigzagging microfeatures.

## Assessing SGN ability to remain in the microfeature around a turn

Since the neurons encountered turn challenges of various microfeature amplitudes, the tendency of these neurons to keep their shaft in the microfeature around a turn was assessed, as previously described [40]. For this, the SGNs which successfully turned in response to the topographical microfeature in the analysis above (Manual assessment of SGN neurite pathfinding) were further studied. The proportion of neurites that remained in the microfeature

around the turn was determined and, subsequently, the proportions were statistically compared with chi-squared testing.

### Live DRGN imaging

Neurons growing in response to micropatterned surfaces were imaged live. To do this, rDRGNs were cultured from mice expressing Pirt-GCaMP3, a genetically encoded calcium indicator. These neurons were cultured on multiangled substrates with 4 μm microfeatures for 18 h after replating. At that timepoint, neurites that were currently growing in a microfeature, actively engaging with the ridge of the feature, and were facing a turn challenge were selected randomly and imaged at a frequency of 15 s for 1.5 h to capture the growth dynamics. These videos were assessed by tracking the growth cones for the duration of the recording and then scored as either turn, exit, or stall based on how they grew for the 1.5 h.

## Results

### Basal $Ca^{2+}$ level in growth cones increases on patterned substrate via IP3 and RyR signaling

Neurons were grown on a topographically patterned substrate consisting of alternating gradually sloped ridges and troughs with an amplitude of 3μm and periodicity of 10μm, as previously described [23, 32, 40]. This system was utilized since it is assumed that the growth cone is always in contact with the ridges of the microfeature. Thus, whatever signaling is utilized by the neuron to sense and grow in response to the topographical features ought to be constitutively active. Replated DRGNs were harvested from Pirt-GCaMP3 transgenic mice and plated on these substrates, as well as on unpatterned substrates as a control, and imaged live to assess the $Ca^{2+}$ level in their growth cones (Fig 1A–1C). In the untreated conditions, the mean normalized $Ca^{2+}$ level is elevated by approximately 10% in the growth cone as the neurites are navigating the narrow topographical microfeatures (Fig 1D). However, if RyR signaling (with 100μM ryanodine) or IP3 signaling (2μM xestospongin C) is inhibited, the $Ca^{2+}$ level in the growth cones of the rDRGNs growing on the substrate does not increase in the neurons growing on the patterned substrate (Fig 1D). This finding implies that topographical patterns increase $Ca^{2+}$ levels by inducing $Ca^{2+}$ release from internal stores via RyR and IP3 signaling.

### Growth cone morphology on topographically patterned substrates is altered when inhibiting $Ca^{2+}$ release from internal stores

Next, the shape of the growth cone as it navigates these features was studied in order to better understand how a rDRGN growth cone alters its morphology in response to topographical growth cues. In untreated growth cones on these patterned substrates (Fig 2A), the shape of a growth cone is more elongated and prolate relative to growth cones on unpatterned surfaces (Fig 2D) [40]. Additionally, the angle between the major axis of the growth cone and the neurite shaft is aligned (Fig 2E). However, when RyR signaling (Fig 2B) or IP3 signaling (Fig 2C) is inhibited in neurons growing on the patterned substrates, the treated growth cones become less prolate in shape (Fig 2D). Furthermore, the treated neurons display a near random orientation of their growth cones relative to their neurite shafts (closer to 45°) (Fig 2E). This observation, when combined with the observations in Fig 1, implies that these two mechanisms of $Ca^{2+}$ release are necessary for growth cones to change their morphology and growth trajectory as the growth cones navigate in response to topographical cues.

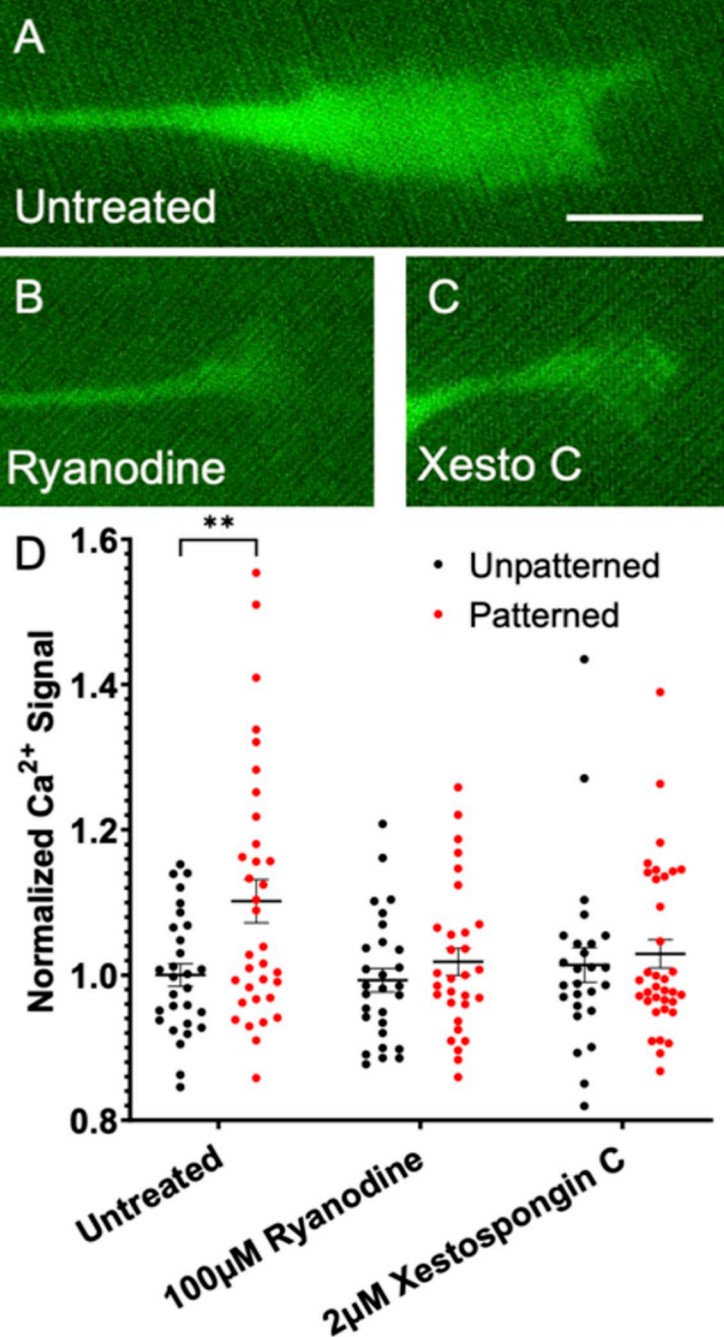

**Fig 1. Basal Ca$^{2+}$ level in growth cones increases on patterned substrate via IP3 and RyR signaling.** (A-C) Representative images of rDRGN growth cones expressing Pirt-GcaMP3 growing on topographically patterned substrates either untreated (A), treated with 100μM ryanodine (B), or 2μM xestospongin C (C). (D) Ca$^{2+}$ signal in the growth cones which is background corrected and normalized to the unpatterned untreated condition. Two-way ANOVA with follow-up Dunnett's multiple comparison testing suggests that the patterned untreated condition is the only significantly different condition ($p < 0.01$). n = 26–35 growth cones per condition from three independent culture replicates from one neonatal mouse. Error bars represent +/-SEM. Scale bar = 5μm.

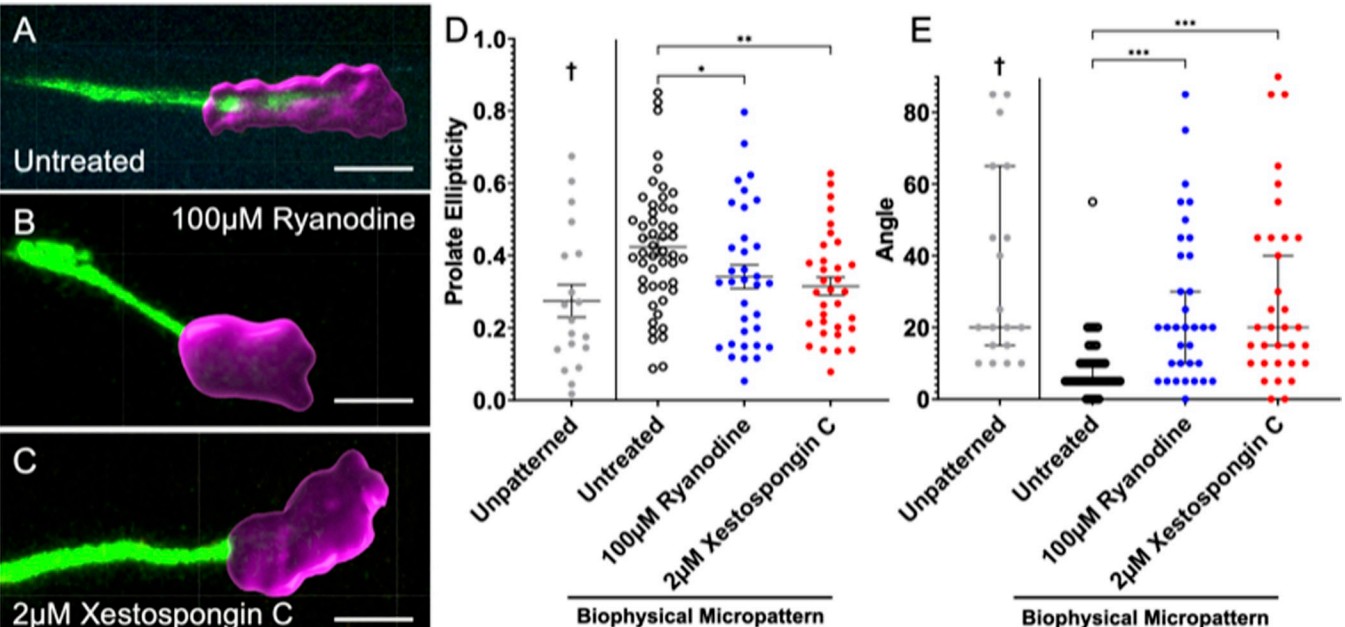

**Fig 2. Inhibiting Ca$^{2+}$ release from internal stores alters growth cone morphology on topographically patterned substrates.** (A-C). Representative images of fixed rDRGN growth cones growing on topographically patterned substrates either untreated (A), treated with 100μM ryanodine (B), or 2μM xestospongin C (C). Growth cone (magenta) was selected and its shape described using Imaris. (D). Growth cone shape was approximated as a spheroid and its prolate ellipticity was calculated. One-way ANOVA with follow-up Dunnett's multiple comparisons testing shows that untreated growth cones on the patterned substrate had greater prolate ellipticity values. Error bars represent +/-SEM, $p < 0.05$. E. The major axis of this spheroid was found, and the angle difference between this axis and the neurite shaft was measured to the nearest 5°. Kruskal-Wallis test with follow-up Dunn's multiple comparisons testing shows that this angle was smaller for the untreated neurons on the patterned substrate. Error bars represent 95% confidence interval. n = 50, 34, 33 growth cones, respectively derived from three independent culture replicates from one neonatal mouse. $p < 0.001$. Scale bars = 5 μm. † indicates dataset from previously published study.

## SGNs guidance to chemo-permissive laminin stripes depends on RyR and IP3 signaling

Given that IP3 and RyR signaling enable alignment to biophysical cues [17], next we assessed if these signaling pathways are also required for SGNs to align to biochemical cues. To do this, SGNs were cultured on a substrate consisting of laminin stripes 25μm alternating with 25μm of photo-deactivated laminin, thus creating a chemo-permissive growth cue. In this experimental system, SGN neurites strongly align to the laminin stripes with a median Alignment Index of 1.15 (Fig 3A). Inhibition of RyR (Fig 3B) or IP3 signaling (Fig 3C) significantly disrupts SGN neurite alignment to the laminin stripes by significantly increasing the median Alignment Index to 1.28 and 1.30, respectively (Fig 3D). Notably, no difference in total neurite length was measured, thus the alignment differences cannot be explained by growth differences (S2 Fig).

## SGNs guidance to chemo-repulsive EphA4-Fc stripes depends on RyR and IP3 signaling

EphA4-Fc serves as a chemo-repulsive cue to SGN neurite growth *in vivo* and *in vitro* [29, 30, 32]. To explore the role of internal Ca$^{2+}$ release to chemo-repulsive SGN neurite guidance, SGNs were cultured on a substrate consisting of EphA4-Fc stripes, repeating every 25μm, adsorbed on a uniform laminin coating [28]. In this experimental system, SGN neurites moderately align to the EphA4-Fc stripes with a median Alignment Index of 1.21 (Fig 4A). Similar to with the laminin stripes (Fig 3), inhibition of RyR (Fig 4B) or IP3 signaling (Fig 4C)

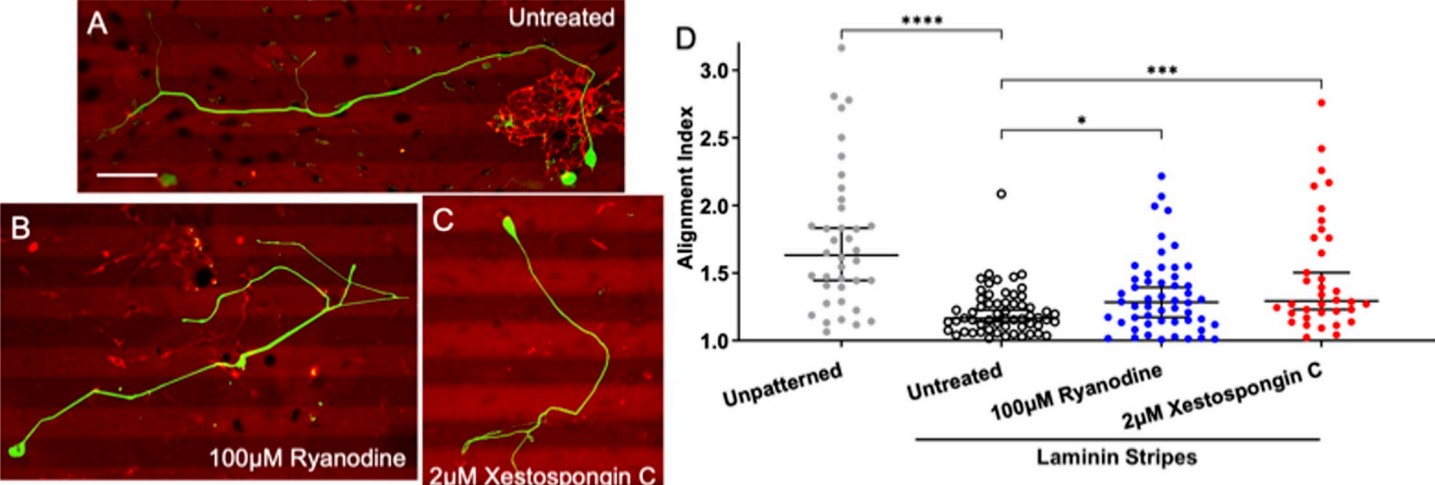

**Fig 3. Inhibiting Ca²⁺ release from internal stores impairs the ability SGNs to follow chemo-permissive laminin stripes.** (A-C). Representative images of fixed SGNs growing in laminin stripe patterned substrates either untreated (A), treated with 100μM ryanodine (B), or 2μM xestospongin C (C) NF200 is labeled green and laminin red. (D) Alignment Index (Total neurite length divided by length in horizontal direction) of SGNs to the micropatterned laminin substrates. Kruskal-Wallis with follow-up Dunn's multiple comparisons testing shows that both treatments impair the ability of SGNs to align to the laminin stripes. Graph shows median Alignment Index +/- 95% CI, p < 0.05. n = 38, 59, 52, 36 neurons derived from three independent culture replicates of a pool of spiral ganglia harvested from multiple litter mates. Scale bar = 50 μm.

significantly disrupts SGN neurite alignment to EphA4-Fc stripes to a median Alignment Index of 1.37 and 1.33, respectively (Fig 4D). Importantly, while there were observed differences in neurite alignment, no difference in total neurite length was measured (S3 Fig).

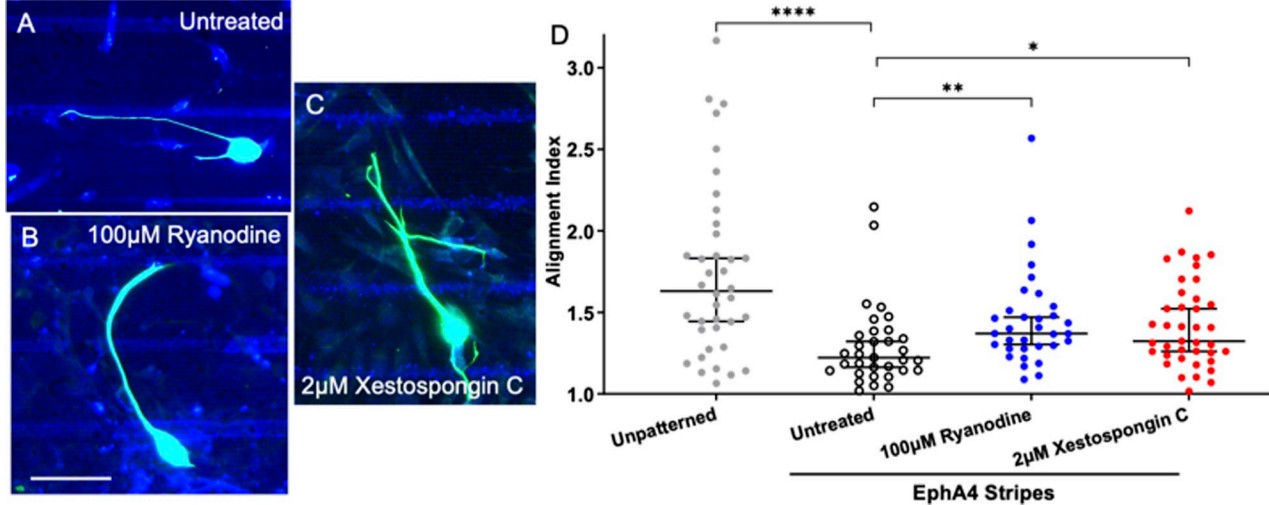

**Fig 4. Inhibiting Ca²⁺ release from internal stores impairs the ability SGNs to follow chemo-repulsive EphA4 stripes.** (A-C). Representative images of fixed SGNs growing in on EphA4-Fc stripe patterned substrates either untreated (A), treated with 100μM ryanodine (B), or 2μM xestospongin C (C). NF200 is labeled aqua, laminin labeled pink, and EphA4-Fc blue. (D) Alignment Index (Total neurite length divided by length in horizontal direction) of SGNs to the micropatterned EphA4-Fc substrates. Kruskal-Wallis with follow-up Dunn's multiple comparisons testing shows that both treatments impair the ability of SGNs to align to the EphA4-Fc stripes. Graph shows median Alignment Index +/- 95% CI, p < 0.05. n = 38, 33, 33, 39 neurons derived from three independent culture replicates of a pool of spiral ganglia harvested from multiple litter mates. Scale bar = 50 μm.

## Inhibiting RyR or IP3 signaling causes two different impaired guidance phenotypes in SGNs growing on to zigzag biophysical micropattern

Given that inhibition of RyR or IP3 signaling impairs SGN alignment to unidirectional topographically and biochemically patterned substrates, next guidance in response to multidirectional cues was assessed. To accomplish this, a topographically patterned substrate consisting of alternating gradually sloped ridges and troughs was engineered using a photomask with zigzagged bands of shaded and unshaded regions [25]. The geometry of the substrate is as follows: amplitude of 4μm, periodicity of 50μm, angle of zigzag is 90°, straight segment of zigzag is 75μm. As this patterned substrate presents a more complex geometry, different and more relevant metrics are needed to analyze SGN growth behavior. A first approach is to trace the neurites, divide the data into 10μm length segments, and then calculate and bin the angle relative to the horizontal direction for each segment. In this, untreated SGNs (Fig 5C) grown on the zigzag pattern tended to have segments aligned to the horizontal and then a gradual decay of proportion as the angle relative to the horizontal is increased (Fig 5F). When treated with either ryanodine (Fig 5D) or xestospongin C (Fig 5E), this distribution becomes less pronounced and more random. However, a more detailed assessment of the treatment groups reveals some nuanced differences. Looking at the xestospongin C treated SGN segment angle data, a minor, but notable, increase is present in the intermediate angles. This is represented by neurites in xestospongin C treated cultures having an increased frequency of segments near 45°, the angle of each segment of the zigzag pattern (Fig 5G). An additional method is to determine the proportion of neurite length that is within the trough of the microfeature. For this measurement, even though the xestospongin C treated SGNs have an altered distribution of angle segments,

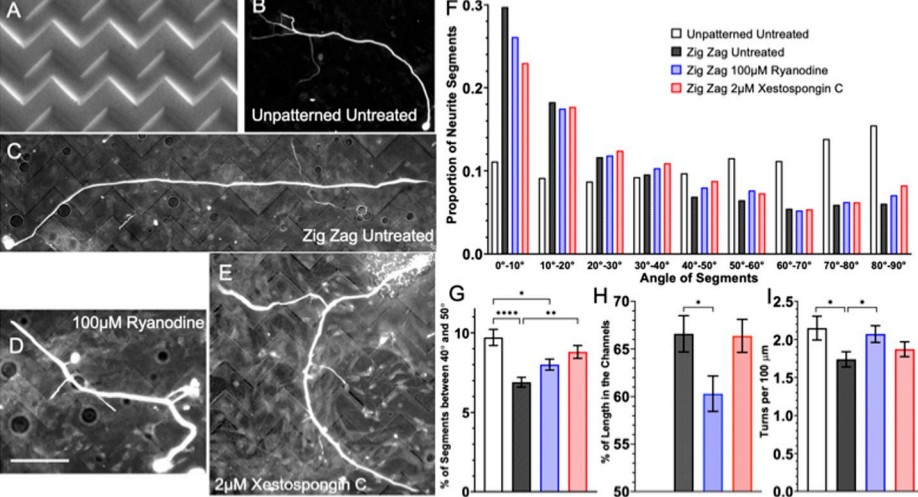

**Fig 5. Inhibiting RyR or IP3 signaling causes different impairment of neurite guidance on a zigzag micropattern.**
(A) SEM image of the topographically micropatterned zigzag substrate. (B) Representative image of fixed SGNs growing on unpatterned substrate.(C-E). Representative images of fixed SGNs growing on topographically micropatterned zigzag substrate either untreated (C), treated with 100μM ryanodine (D), or 2μM xestospongin C (E). (F) Distribution of SGN neurite segment angles relative to the horizontal axis. (G) Proportion of neurite segments which are oriented between 40° and 50° relative to the horizontal, i.e., the angle of each segment of the zigzag micropattern. Error bars are +/- standard error of proportion. Significance indicated for chi-square test with follow-up z-testing. p < 0.01. (H) Average percent of length for each neuron that is in the micropatterned feature. 50% would represent a random tendency for the neuron. (I) Average number of turns measured per neuron normalized by 100 μm of length. For H & I, one-way ANOVA with follow-up Dunnett's multiple comparisons testing was done to compare groups. Error bars are +/- SEM, p < 0.05. n = 53, 100, 109, 89 neurons derived from three independent culture replicates of a pool of spiral ganglia harvested from multiple litter mates. Scale bar = 50 μm.

the same proportion of the total neurite shaft length is present in the trough of the zigzag microfeature (Fig 5H). This observation contrasts with the neurite growth behavior in the ryanodine treated SGNs which appears to approximate SGNs on unpatterned substrates, suggesting a more general impaired alignment to these multidirectional cues. First, the ryanodine treated neurons exhibit a significantly smaller proportion of their total length in the microfeature trough (Fig 5H) compared to untreated neurons on these patterned surfaces. Additionally, the number of turns a neurite makes can be calculated and, similar to neurons grown on unpatterned substrates, the ryanodine treated neurons display a greater number of turns per unit length than untreated neurons on the zigzag pattern (Fig 5I). Lastly, no difference in total neurite length was observed between the treatment groups (S4 Fig).

## Ryanodine treatment inhibits the ability of SGN neurites to navigate complex biophysical turn challenges, while xestospongin C treatment promotes aligning across the angled feature

These observations imply distinct roles of these two signaling pathways in neurite guidance and prompted further research into how RyR and IP3 signaling impact SGN neurite guidance to more complex geometry topographical cues. Thus, a biophysical substrate was engineered consisting of six different angled microfeature turn challenges, ranging from 30–180 (straight) degrees [40], where the angled channels are not overlapping with each other as occurs with the zigzag system pattern (Fig 5A). Neurite pathfinding in response to this system is complex and the effect size is small, due in part to the fact that the pattern only occupies a small percentage of the overall surface area of the substrate. These limitations are offset by the substrate facilitating analyses of growth cone and neurite behavior at specific topographical microfeatures. A first method to quantify the SGN behavior in response to these angled turns is to measure the length each neurite navigates in the microfeatures. This was done by tracing the length from the point where the neurite entered the microfeature to where it either exited the microfeature or the endpoint of the neurite was reached. In this metric, untreated SGN neurites (S5A Fig) track inside the microfeatures significantly longer than both the ryanodine (S5B Fig) and xestospongin C (S5C Fig) treated groups. In particular, there was approximately a 20% reduction in median tracking in both treatment groups (S5D Fig).

Additionally, this same experimental system was used to more thoroughly assess neurite behavior by employing another method to quantitatively score each neurite encounter with an angled microfeature. In this other analysis, neurites were classified as (1) does not follow (Fig 6A), (2) makes a turn (Fig 6B), or (3) aligns across the zigzag (Fig 6C). Notably, classifications (2) and (3) were both deemed to be behaviors of neurites following an angled guidance cue. The proportions were plotted as a stacked bar graph for each angle turn and treatment group (Fig 6D). Interestingly in this assessment, xestospongin C treated SGNs follow the microfeature turns to the same extent as untreated SGNs. In particular, xestospongin C treated SGNs are observed frequently aligning across the zigzag in the sharp angle turn conditions (Fig 6C and S5C Fig). Significantly, the proportion of neurites that remain in the channel around the turn are reduced in the xestospongin C treated group (Fig 6E).

## Real time analysis of growth cone behavior suggest role of RyR signaling in growth cone repulsion and IP3 signaling in turning

Given the challenges and limitations in analyzing neurite guidance using fixed cultures in complex topographically patterned systems, next rDRGNs from Pirt-GCaMP3 transgenic mice were cultured and imaged in real time for 1.5h as they grew on micropatterned substrate. The features in these micropatterns were engineered with moderate amplitudes (4 μm), so that

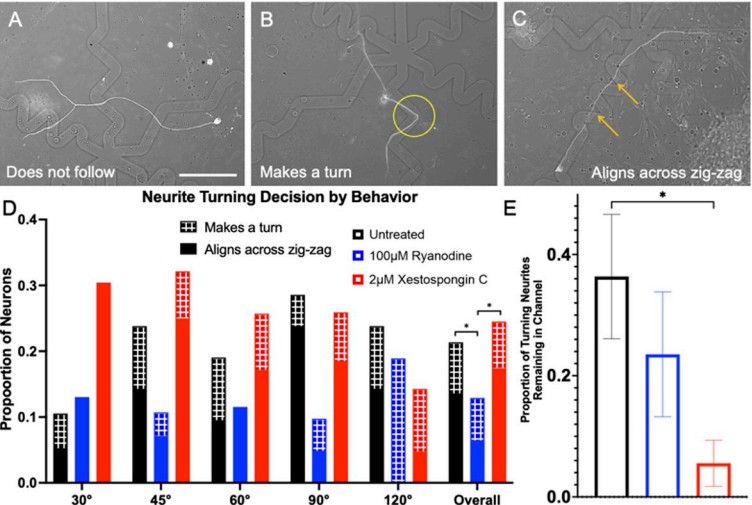

**Fig 6. Xestospongin C treated SGNs follow the turn challenges, but do not remain in the microfeature.** (A-C). Representative images of fixed SGNs growing in response to topographical microfeature turn challenges. (A) SGN not following the microfeatures (B) SGN making a turn and holding its position in a 90˚ feature turn, (C) SGN aligning across the 45˚ turn microfeature. (D) Proportion of neurites which were observed to follow microfeature turns by angle and drug treatment. Patterned segment of a bar is the proportion which made one turn, while solid demonstrated alignment across multiple turns. Multinomial logistic regression shows effect of Ryanodine treatment and microfeature angle on guidance. Measurements in each treatment were compiled and results from Chi-square test with follow up z-tests shown for overall data. Error bars represent +/- standard error for a proportion. n = 103, 155, 155. (E) Proportion of the neurites that made a turn and held its position in the microfeature. Chi-squared test with follow up z-tests indicate a lower proportion of xestospongin C treated remain in the channel during a turn. Error bars +/- SEM, p < 0.05. n = 22, 17, 36 neurite shafts from three independent culture replicates of a pool of spiral ganglia harvested from multiple litter mates. Scale bar = 50 μm.

neurites would be responsive but not constrained by the microstructure. In the videos, the untreated growth cones were highly dynamic and reacted to the microfeature ridges in the 4μm amplitude condition (Fig 7A and 7B). In this dynamic system, growth cones were imaged as they encountered a micropattern feature and their behavior scored as turn, stall, or exit. In untreated growth cones(Fig 7A and 7B), there is roughly an even distribution of behavior by turning, stalling, or exiting, 27%, 40%, and 33% respectively. Ryanodine treatment led to a strong decrease in the stalling behavior (0%) with an increase in exiting (80%) behavior (Fig 7C and 7D). Conversely in the xestospongin C treated cultures (Fig 7E and 7F), there is a modest decrease in the proportion turning (12.5%) and an increase in the proportion stalling (50%) (Fig 7G).

## Discussion

Medical device and bioengineers aspire to improve the interface between neural prosthetic electrodes and their target neurons. In this pursuit, they seek to engineer biomaterial surfaces or coatings of the electrode to guide neurite outgrowth into close proximity to or contact with the electrode array [5, 41–43]. In regard to CIs, neurotrophic molecule gradients, patterned biochemical cues, and topographical surface features have been explored as methods to steer SGN neurite outgrowth towards CI electrodes [44–47]. Based on prior work showing its key role in neurite pathfinding, understanding the role of $Ca^{2+}$ signaling in facilitating neurite guidance in response to varied growth cues helps in efforts to accurately orchestrate this process. Thus, in this study we provide compelling evidence on the role of two sources of $Ca^{2+}$

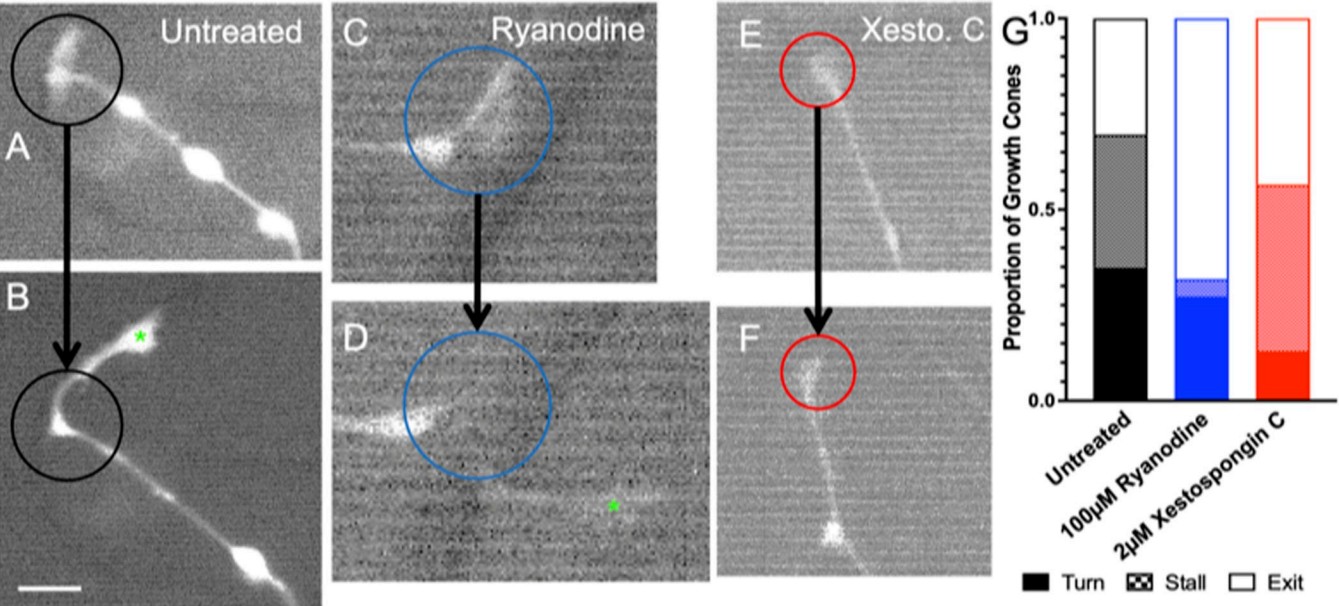

**Fig 7. Ryanodine treated growth cones do not respond to the biophysical microfeatures while xestospongin C treated turn at a decreased frequency.** (A, B) Before and after images of an untreated growth cone navigating a turn. Black circle shows same ROI in each and new growth is marked by green asterisk in B. (C, D) Before and after images of an 100μM ryanodine treated growth cone that exited the microfeatures after encountering a turn. Blue circle shows same ROI in each and new growth is marked by green asterisk in D. (E, F). Before and after images of an 2μM xestospongin C treated growth cone that stalled after encountering a turn. Red circle shows same ROI in each. G. Frequency comparison of turning behaviors of rDRGN growth cones while navigating 4μm amplitude. Chi-squared test with follow up z-tests indicate a lower proportion of ryanodine treated neurite display stalling and xestospongin C treated turn. n = 22, 22, 23 growth cones derived from six independent culture replicates from one neonatal mouse. Scale bar = 10 μm.

release from internal stores, IP3 and RyR, and their respective roles in how growth cones of sensory neurons navigate in response to biochemical and biophysical cues.

## IP3 and RyR act in the growth cone when neurites are navigating biophysical cues

As prior work has demonstrated, these signaling pathways are required for neurites to align their growth in response to topographical microfeatures [17], and IP3 signaling occurs in the growth cone during pathfinding to chemo-attractive cues [15, 16]. Thus, we first assessed the role of RyR and IP3 signaling in the growth cone during pathfinding in response to topographical guidance cues. First, there is typically an increase in growth cone $Ca^{2+}$ as neurites sense and pathfind in response to these biophysical cues. However, if either IP3 or RyR signaling is inhibited, this increase in growth cone $Ca^{2+}$ is blunted (Fig 1).

These findings suggest the presence of RyR and IP3 signaling pathways in the growth cone during neurite pathfinding in response to biophysical cues. To further explore their contribution to SGN neurite pathfinding we assessed growth cone morphology on the micropatterned substrates in the absence or presence of inhibitors. Typically, when grown on parallel rows of repeating microfeatures of ridges and troughs, the growth cone becomes prolate in shape and its long axis is aligned with the neurite shaft [40]. However, if IP3 or RyR signaling is inhibited, the growth cone become less prolate in shape (Fig 2D) and the growth cone and neurite shaft become less aligned in their orientation (Fig 2E). Combining these observations, it appears that growth cones engage both RyR and IP3 signaling pathways as they sense and grow in response to biophysical guidance cues.

## IP3 and RyR signaling are also integral to SGN pathfinding in response to biochemical cues

Next, we assessed the role of these signaling pathways in SGN pathfinding in response to micropatterned biochemical substrates. Prior work has shown that growth cones recruit similar signaling mechanisms in response to these biophysical cues as chemo-repulsive cues. For example, biophysical and chemo-repulsive cues activate RhoA and Rho associated kinase to mediate growth cone guidance [11, 48]. Likewise, elevation of cyclic adenosine monophosphate enables SGN growth cones to grow beyond biophysical and chemo-repulsive borders [17, 49]. Additionally, others have shown differential roles of IP3 and RyR signaling in turning in response to chemo-attractive cues [15]. Thus, we engineered two different biochemically micropatterned substrates: chemo-permissive laminin stripes [27] and chemo-repulsive EphA4-Fc stripes [28, 32]. First in the chemo-permissive system SGN neurites strongly track the laminin stripe, however, when IP3 or RyR signaling was inhibited the neurites tended to wander and not be aligned as strongly in the direction of the stripes (Fig 3).

In the second system, SGN neurites avoid the EphA4-Fc stripes. Similar to the chemo-permissive system, treatment with ryanodine or xestospongin C impaired neurite guidance in response to these chemo-repulsive stripes (Fig 4). Together the data suggest that the function of these two signaling pathways extend beyond being secondary messengers for mechanosensitive channels in responding to biophysical cues. $Ca^{2+}$ release from internal stores appears to contribute to growth cones ability to sense and pathfind in response to biochemical cues as well.

## Neurite pathfinding to multi-directional micropatterns and turn challenges cues reveals differential roles of IP3 and RyR signaling

While the data here suggest that IP3 and RyR signaling both contribute to growth cone pathfinding, work by others has identified situations where they function differently in neurite guidance [14, 15]. Therefore, we sought to study the role of these signaling in pathfinding to more complex multidirectional microfeatures to ascertain if they have distinct roles in the orchestrated process of a growth cone sensing a repulsive cue, halting growth, coordinating a turn, and growing in a new direction.

A first system to test this is culturing SGNs on repeating zigzagging microfeatures with 90° turns [25]. Normally on this substrate, the neurites align in the horizontal direction of the zigzags and tend to somewhat track the microfeatures (~2/3 of neurite length remains in the features). We see that SGN neurites treated with either ryanodine or xestospongin C display impaired guidance to these zigzags micropatterns (Fig 5C–5F). Further assessment of the behavior of the neurites on this system suggests differential behavior of the treatment groups. First, when inhibiting RyR signaling, the neurites display a generalized shift in behavior towards that expected with neurons growing on an unpatterned substrate: modest shift in segment orientation towards randomness (Fig 5F), neurites spend less time in the microfeature trough (Fig 5H), and the neurites tend to turn or wander more frequently (Fig 5I). Overall, this difference implies a general inhibition of the ability of the growth to pathfind in response to the multidirectional biophysical cues with RyR inhibition. However, IP3 signaling inhibition results in a different pathfinding behavior, as the neurites follow the zigzags in a more subtle way. Neurites treated with xestospongin C still respond to the multidirectional microfeatures as they have the same proportion of their length in the trough as the untreated condition (Fig 5H). However, neurites in cultures treated with Xestospongin C demonstrate a higher proportion of their length aligned in the diagonal direction of zigzag (Fig 5G). It is challenging to determine the specific effect that IP3 signaling plays in neurite pathfinding using

these data alone, prompting the need for further study of growth cones sensing and turning in response to other cues of complex geometries.

Thus, in the next system we assessed the neurite's ability to track and turn in response to various microfeature turn challenges [40]. In the system engineered herein, the micropatterned features present angles ranging from 30–180 degrees and comprise only a small percentage of the overall surface area. In 4 μm amplitude microfeatures, untreated neurites do not follow this system very strongly due to the random nature in which the neurites encounter the turn challenges. However, similar to the multidirectional micropatterned system above, the contribution of RyR and IP3 signaling becomes clearer with further assessment of neurite behavior. A first measurement of how well a neurite follows these turns is to measure how long the neurite remains in the microfeatures. In this, we see a significant reduction in length followed in both the ryanodine and xestospongin C treatment groups (S5 Fig). Next, we categorically assessed how well the neurites turned in response to the microfeatures using both fixed (Fig 6) and real time systems (Fig 7). RyR inhibition blunted the ability of neurites to navigate these biophysical cues manifesting as a reduction in the ability of the neurites to turn and follow the cues in the fixed system (Fig 6) as well as a reduction in engagement with the microfeature ridge during live imaging (Fig 7). Together these findings suggest an overall impairment in sensing and responding to the biophysical features when RyR is inhibited.

In assessing the ability of the xestospongin C treated neurites to turn in response to the micropatterns, an interesting trend emerges. First in the fixed system, xestospongin C treated SGNs follow the microfeature turns as well as the untreated SGNs. In particular, an increased proportion of the neurites align across the zigzag (Fig 6C and 6D) in the sharp angle turn challenges (30˚ and 45˚). These neurites remain in the microfeature trough at much reduced rate compared to untreated cultures (Fig 6E). This behavior implies that inhibition of IP3 signaling results in a greater number of neurites that grow straight across the angled features. This observation is consistent with what is seen with these same neurites in live imaging. First, IP3 inhibition reduces neurites turning in response to the features and importantly, also increases the proportion of neurites that stall at the feature edge (Fig 7). The predisposition of neurites treated with Xestospongin C to cut across the zigzagging turning features coupled with their stalling at feature edges suggests that IP3 signaling enables growth cones to coordinate turning behavior in response to microtopographical features, i.e. when IP3 signaling is inhibited growth cones appear to sense the ridge, stall, fail to coordinate a turn, and then grow straight across the microfeature.

## Integrating into a model of neurite guidance to repulsive substrate cues

A significant body of work has revealed the role of $Ca^{2+}$ and IP3 signals in integrating growth cone responses to guidance cues. Prior to integrating a model of growth cone guidance to the patterned substrates used here, it is important to understand the characteristics of the diverse cues used in the various experimental systems in previous studies and the $Ca^{2+}$ signals they induce. Importantly, the guidance cues engineered here differ from instantaneous chemical gradient perfusion or microinjections into the cytoplasm. These patterned substrates are designed such that the growth cones encounter robust and consistent growth cues. Extra- or intracellular injections result in transient changes to $Ca^{2+}$ signaling [15, 18, 50, 51], while in the topographically patterned substrates the growth cones interact with a consistent cue through time and space. This is important since neurite guidance is governed by both the temporal and spatial aspects of $Ca^{2+}$ signaling [13, 14, 52]. Further, various attractive cues induce different signaling pathways in the growth cone [15, 51]. These nuanced differences in methodologies combined with the intricacies of the tightly choreographed role of $Ca^{2+}$ level in

growth cone pathfinding should be appreciated to help integrate these data here with other presented in the field.

Research into the role of RyR signaling, $Ca^{2+}$ induced $Ca^{2+}$ release (CICR), and $Ca^{2+}$ elevation in neurite guidance likewise presents a complex picture. For example, the findings range from RyR being involved in sensing repulsive cues [17], to CICR via RyR acting as a crucial mediator for attractive turning [18, 51]. Furthermore, asymmetrical increases in $Ca^{2+}$ in the growth cone arising from guidance cues present additional nuance; small elevations in $Ca^{2+}$ via extracellular sources as well as large or global increases in $Ca^{2+}$ both lead to repulsion, whereas moderate, asymmetrical elevations in $Ca^{2+}$ levels induce attraction [13, 52].

With these nuances in mind, we can integrate the data presented here into a model of the role of $Ca^{2+}$ and IP3 signaling in the growth cone as neurites navigate biophysical or biochemical micropatterned features. Prior work suggests that topographical microfeatures function as a repulsive cue and recruit similar downstream signaling as chemo-repulsive molecules including RhoA and Rho associated kinase [11]. In the model presented here, the growth cone initially senses the repulsive cue and initiates a local increase in $Ca^{2+}$ (repulsive signal) though plasma membrane sensors and channels [11, 32, 53–56]. This localized $Ca^{2+}$ elevation results in CICR via RyRs (Fig 8A). Importantly, this CICR is expected to be sufficiently robust in magnitude, spatially and temporally, to stall growth extension localized to region of the growth cone interacting with the microfeature (e.g., ridge or chemo-repulsive border) [9, 57].

For neurite turning to a repulsive cue, if the growth cone encounters the repulsive cue at an angle, it is suspected that RyR-mediated CICR is localized to the region of growth cone interacting with the feature. The other side of the growth cone, engaging a permissive substrate (e.g., biophysical groove or laminin coating), would extend via IP3 signaling induced attraction [15, 58–61] (Fig 8B). IP3 signaling has been shown to harmonize turning in response to chemo-attractive cues, such as a gradient of diffusible attractive cues (e.g., neurotrophins) [15, 17] whereby IP3 signaling coordinates asymmetrical $Ca^{2+}$ signaling in the growth cone [16]. This spatial asymmetry in RyR-mediated CICR and IP3 signaling would lead to the coordinated turning observed. In combination, the findings suggest that RyR-mediated CICR and IP3 signaling play distinct roles in coordinating neurite pathfinding. Their roles combine to orchestrate growth cone guidance in response to the cues studied here.

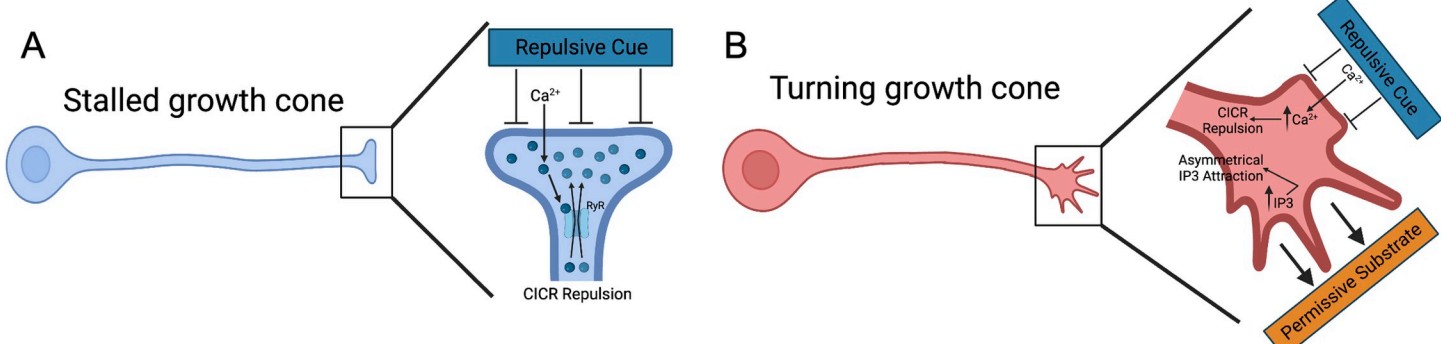

**Fig 8. Model of RyR and IP3 signaling in neurite guidance to patterned substrate cues.** A. A stalled neurite whereby a repulsive cue initiates $Ca^{2+}$ influx via yet to be identified plasma membrane sensors and channels, which may vary for each type of repulsive cue. This local increase in $Ca^{2+}$ leads to RyR mediated CICR and repulsion. B. A turning neurite whereby the phenomenon in Fig 8A is occurring in the region being turning away from. While in the region being turned toward, the growth cone is sensing attractive cues on the permissive substrate, which triggers IP3 signaling and attractive growth.

## Conclusions

Here we use a variety of engineered micropatterned substrates to probe the role of IP3 and RyR signaling for sensory neuron neurite pathfinding. Our data suggest that these mechanisms for $Ca^{2+}$ release from internal stores play essential and distinct roles in the process. First, RyR signaling appears critical for growth cones to halt growth upon encountering patterned, repulsive substrate cues. Meanwhile, IP3 signaling appears necessary for coordinated growth cone turning. Taken together, these data highlight the overlap in the intracellular signaling processes governing neurite guidance in response to a diversity of cues and motivates further work on the role of these pathways in regenerative pathfinding using *in vivo* models.

## Supporting information

**S1 Fig. Methods to measure neurite behavior in response to angled topographical microfeature.** (A) Schematic demonstrating the derivation of percent of neurite length in topographical microfeature. (B) Schematic demonstrating calculation of turns. Neurite is segmented in to 10 μm segments and the angle relative to the horizontal calculated for each segment. The MATLAB program counts a turn when 3 consecutive segments vary by >10˚. (C) Schematic demonstrating the measurement of the length a neurite remains in topographical microfeature. Scale bars = 50 μm.
(TIF)

**S2 Fig. Inhibiting $Ca^{2+}$ release from internal has no effect on SGN neurite length on chemo-permissive laminin stripes.** Total neurite length of SGNs cultured in the respective conditions. Kruskal-Wallis testing shows no effect of treatment on SGN neurite growth. Graph shows median Alignment Index +/- 95% CI. n = 38, 59, 52, 36 neurons.
(TIF)

**S3 Fig. Inhibiting $Ca^{2+}$ release from internal has no effect on SGN neurite length on chemo-repulsive EphA4 stripes.** Total neurite length of SGNs cultured in the respective conditions. Kruskal-Wallis testing shows no effect of treatment on SGN neurite growth. Graph shows median Alignment Index +/- 95% CI. n = 38, 33, 33, 39 neurons.
(TIF)

**S4 Fig. Inhibiting RyR or IP3 signaling has no effect on neurite length on a zig zag Micropatterned Substrate.** Total neurite length of SGNs cultured in the respective conditions. Kruskal-Wallis testing shows no effect of treatment on SGN neurite growth. Graph shows median Alignment Index +/- 95% CI. n = 53, 100, 109, 89 neurons.
(TIF)

**S5 Fig. Inhibiting $Ca^{2+}$ release from internal stores impairs guidance to angled biophysical microfeatures.** (A-C). Representative images of fixed SGNs growing in response to topographical microfeature turn challenges either untreated (A), treated with 100μM ryanodine (B), or 2μM xestospongin C (C). (D) Length neurite remains in topographical microfeature once encountering it. Two-way ANOVA on ranks suggests that both treatments impair the ability of SGNs to follow the topographical microfeature turn challenges and length followed increases with more gradual turns. Data are 95% confidence interval box and whisker. n = 29, 645, 355, 348 neurite encounters. Scale bar = 100 μm.
(TIF)

**S1 Data. Raw data corresponding to Figs 5, 6D and S4.**
(XLSX)

**S1 File. Additional description of statistics and data visualization.**
(PDF)

## Acknowledgments

The Pirt-GCaMP3 transgenic mice were gifted from Dr. Xinzhong Dong of John Hopkins University.

## Author Contributions

**Conceptualization:** Joseph T. Vecchi, Marlan R. Hansen.

**Formal analysis:** Joseph T. Vecchi.

**Funding acquisition:** Joseph T. Vecchi, C. Allan Guymon, Marlan R. Hansen.

**Investigation:** Joseph T. Vecchi, Madeline Rhomberg.

**Methodology:** Joseph T. Vecchi, Madeline Rhomberg.

**Project administration:** C. Allan Guymon, Marlan R. Hansen.

**Resources:** C. Allan Guymon, Marlan R. Hansen.

**Software:** Joseph T. Vecchi.

**Supervision:** C. Allan Guymon, Marlan R. Hansen.

**Validation:** Joseph T. Vecchi, Madeline Rhomberg.

**Visualization:** Joseph T. Vecchi.

**Writing – original draft:** Joseph T. Vecchi.

**Writing – review & editing:** Joseph T. Vecchi, Madeline Rhomberg, C. Allan Guymon, Marlan R. Hansen.

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
