## [Decision Letter · Decision Letter 0]

1 May 2024

PONE-D-24-12078Inositol trisphosphate and ryanodine receptor signaling distinctly regulate neurite pathfinding in response to engineered micropatterned surfacesPLOS ONE

Dear Dr. Vecchi,

Thank you for submitting your manuscript to PLOS ONE. After careful consideration, we feel that it has merit but does not fully meet PLOS ONE’s publication criteria as it currently stands. Therefore, we invite you to submit a revised version of the manuscript that addresses the points raised during the review process.

We look forward to receiving your revised manuscript.

Kind regards,

Sujeong Jang

Academic Editor

PLOS ONE

Journal Requirements:

   "MH and AG: NIH-NIDCD R01-DC012578

JV: NIH-NIDCD F31-DC020371, and University of Iowa: NIGMS T32-GM007337" 

5. We note that your Data Availability Statement is currently as follows: "All relevant data are within the manuscript and its Supporting Information files."

Additional Editor Comments:

Thank you for submitting your study on PLOS ONE.

This study presents a comprehensive investigation into the role of IP3 and RyR signaling in neurite pathfinding of sensory neurons in response to biophysical cues in vitro. We deeply reviewed with 2 other reviewers, and decided for major revision.

Reviewers' comments:

Reviewer's Responses to Questions

**Comments to the Author**

1. Is the manuscript technically sound, and do the data support the conclusions?

Reviewer #1: Yes

Reviewer #2: Yes

2. Has the statistical analysis been performed appropriately and rigorously? 

Reviewer #1: Yes

Reviewer #2: I Don't Know

3. Have the authors made all data underlying the findings in their manuscript fully available?

Reviewer #1: Yes

Reviewer #2: Yes

4. Is the manuscript presented in an intelligible fashion and written in standard English?

Reviewer #1: Yes

Reviewer #2: Yes

5. Review Comments to the Author

Reviewer #1: Results section – make findings clearer and more concise.

Line 21: do you need a – after Micro-? This recurs throughout the paper. Consider removing – on the first word.

-This is repeated at other places in the paper

Line 54: “into close proximity to stimulating” – difficult to read may want to change “to” to

Successfully shows that disrupting Ca signaling alters neurite growth direction.

Line 446: Is this 20% reduction significant?

Good paper highlighting the importance in Ca pathways in direction of neurite growth.

Excellent Figures.

Reviewer #2: This study presents a comprehensive investigation into the role of IP3 and RyR signaling in neurite pathfinding of sensory neurons in response to biophysical cues in vitro. The manuscript is well-written, the conclusions are appropriate and supported by the data, and the findings are likely to be of interest to a wide audience as neurite growth is important for many biological processes and applications. The use of different combinations of micropatterned geometries and biochemical cues illustrates nicely the different functions of IP3 and RyR signaling in neurite pathfinding. The main drawback of this manuscript is that the effect sizes appear to be small and very specific to the measurements chosen, bringing into question the biological significance of the findings. Some clarification on the specific points below will help to address this concern.

1. The introduction provides a good overview of the background into the field and existing technologies for the study of biochemical and biophysical cues for neurite outgrowth. The motivation for the study is generally clear although the authors could revise the paragraph on page 3 beginning on line 81 to more clearly state where the knowledge gap is within Ca2+ signaling and the RyR/IP3 pathways that this study addresses. Mainly, to answer the question of if is it known that they are all involved in neurite guidance, but it is unknown which type of environmental signal they respond to?

2. Page 3, line 60: typo “In this, the growth cone plays a pivotal role in pathfinding responding to diverse…”

3. The authors should provide more rationale for the three micropatterned geometries chosen for this study. Some explanation is peppered through the results and discussion but clearer statements upfront would enhance the understanding of the findings

4. Since almost all experiments involve the use of ryanodine and xestospongin C the authors should include more explanation of how the concentrations were chosen and confirmed to inhibit RyR/IP3 signaling? Is inhibition with these drugs complete? Does administration of the drug affect cell viability?

5. In addition to the number of technical replicates (e.g., growth cones, etc), the biological sample size should be stated in the figure legends – i.e., the number of animals neurons were derived from. Can the authors also comment on how the statistical tests were conducted, i.e., which sample size was used for the analysis?

6. Did the authors confirm that neuron viability is unaffected by culture on micropatterned substrates?

7. Page 6, line 197: please revise sentence so it is clear which drugs are added to cells in these experiments

8. Page 6, line 201 and page 6, line 213: how were growth cones identified and ROIs selected? NF200 expression is not specific to the growth cone

9. Page 9, line 326: what is the biological relevance or significance of changes in growth cone morphology and orientation, and the relation to Ca2+ signaling?

10. Since the analysis method appears to be unique to this manuscript and is somewhat complex to understand with text alone, Section 2.10, 2.11, and 2.12 may benefit from a supplementary figure illustrating the explanations of how neurite turning was classified and scored

11. Page 10, line 369: The authors should provide references for the statement “EphA4-Fc serves as a chemorepulsive cue to SGN neurite growth…”

12. The authors should consider also testing DRGNs in the more complex zigzag topographically patterned substrates to compare to the SGNs. It would be interesting to know if different types of sensory neurons responded differently to drug treatments

13. Can the authors comment on the effect of inhibiting RyR and IP3 together? Would there be an additive inhibitory effect on neurite growth? If those experiments were performed, please include in the manuscript, or if not authors can comment on the hypothesized effect

6. PLOS authors have the option to publish the peer review history of their article (what does this mean?). If published, this will include your full peer review and any attached files.

Reviewer #1: **Yes: **Reid Bartholomew

Reviewer #2: **Yes: **Samantha Payne

---

## [Author Response · Author response to Decision Letter 0]

12 Jun 2024

See response to reviewer document.

---

## [Decision Letter · Decision Letter 1]

26 Jun 2024

Inositol trisphosphate and ryanodine receptor signaling distinctly regulate neurite pathfinding in response to engineered micropatterned surfaces

PONE-D-24-12078R1

Dear Dr. Hansen,

We’re pleased to inform you that your manuscript has been judged scientifically suitable for publication and will be formally accepted for publication once it meets all outstanding technical requirements.

Kind regards,

Sujeong Jang

Academic Editor

PLOS ONE

Additional Editor Comments (optional):

Reviewers' comments:

Reviewer's Responses to Questions

**Comments to the Author**

1. If the authors have adequately addressed your comments raised in a previous round of review and you feel that this manuscript is now acceptable for publication, you may indicate that here to bypass the “Comments to the Author” section, enter your conflict of interest statement in the “Confidential to Editor” section, and submit your "Accept" recommendation.

Reviewer #2: All comments have been addressed

Reviewer #3: All comments have been addressed

2. Is the manuscript technically sound, and do the data support the conclusions?

Reviewer #2: Yes

Reviewer #3: Yes

3. Has the statistical analysis been performed appropriately and rigorously? 

Reviewer #2: Yes

Reviewer #3: Yes

4. Have the authors made all data underlying the findings in their manuscript fully available?

Reviewer #2: Yes

Reviewer #3: Yes

5. Is the manuscript presented in an intelligible fashion and written in standard English?

Reviewer #2: Yes

Reviewer #3: Yes

6. Review Comments to the Author

Reviewer #2: (No Response)

Reviewer #3: All comments have been addressed adequately and I agree that it is suitable for publication at this time.

7. PLOS authors have the option to publish the peer review history of their article (what does this mean?). If published, this will include your full peer review and any attached files.

Reviewer #2: **Yes: **Samantha Payne

Reviewer #3: No

---

## [Editor Report · Acceptance letter]

24 Jul 2024

PONE-D-24-12078R1 

PLOS ONE

Dear Dr. Hansen, 

I'm pleased to inform you that your manuscript has been deemed suitable for publication in PLOS ONE. Congratulations! Your manuscript is now being handed over to our production team.

Kind regards, 

on behalf of

Dr. Sujeong Jang 

Academic Editor

PLOS ONE